# Precision Killing of Sinoporphyrin Sodium-Mediated Photodynamic Therapy against Malignant Tumor Cells

**DOI:** 10.3390/ijms231810561

**Published:** 2022-09-12

**Authors:** Guixiang Lv, Zhihui Dong, Yunhan Zhao, Ning Ma, Xiaochen Jiang, Jia Li, Jinyue Wang, Jiaxin Wang, Wenxiu Zhang, Xin Lin, Zheng Hu

**Affiliations:** 1Department of Biochemistry and Molecular Biology, Harbin Medical University, Harbin 150086, China; 2Laboratory of Sono- and Photo-Theranostic Technologies, Harbin Institute of Technology, Harbin 150080, China

**Keywords:** photodynamic therapy, parameters, outcomes, *e*-exponential function, dosimetry

## Abstract

Photodynamic therapy (PDT) has significant advantages in the treatment of malignant tumors, such as high efficiency, minimal invasion and less side effects, and it can preserve the integrity and quality of the organs. The power density, irradiation time and photosensitizer (PS) concentration are three main parameters that play important roles in killing tumor cells. However, until now, the underlying relationships among them for PDT outcomes have been unclear. In this study, human malignant glioblastoma U-118MG and melanoma A375 cells were selected, and the product of the power density, irradiation time and PS concentration was defined as the total photodynamic parameter (TPP), in order to investigate the mechanisms of PS sinoporphyrin sodium (DVDMS)-mediated PDT (DVDMS-PDT). The results showed that the survival rates of the U-118MG and A375 cells were negatively correlated with the TPP value in the curve, and the correlation exactly filed an *e*-exponential function. Moreover, according to the formula, we realized controllable killing effects of the tumor cells by randomly adjusting the three parameters, and we finally verified the accuracy and repeatability of the formula. In conclusion, the establishment and implementation of a newly functional relationship among the PDT parameters are essential for predicting PDT outcomes and providing personalized precise treatment, and they are contributive to the development of PDT dosimetry.

## 1. Introduction

According to statistics from the World Health Organization (WHO), malignant tumors caused the deaths of nearly 10 million cancer patients worldwide in 2020, becoming the world’s leading cause of death [1]. At present, the current treatment methods for malignant tumors have still not achieved satisfactory outcomes. Therefore, it is urgent to develop new tumor treatment strategies.

Photodynamic therapy (PDT) has become one of the most promising physical therapies in recent years [2,3,4,5]. It mainly adopts light at a specific wavelength to stimulate the photosensitizer (PS) specifically accumulated in the tumor, and the excited PS can transfer energy to oxygen, thereby producing singlet oxygen (^1^O_2_) to kill the tumor cells [4,6]. PDT has the characteristics of good targeting, significant killing effects and less trauma [4,7]. Previous studies have shown that PDT is effective in the treatment of various malignant tumors, and it has been clinically approved for the treatment of glioblastoma (GBM) [8], skin cancer [9], esophageal cancer [10], lung cancer [11], head and neck cancer [12], etc.

In recent years, a growing body of research has shown that the power density, irradiation time and PS concentration are the three most important parameters for PDT to exert its tumor-killing effects [13,14,15,16,17,18]. However, because of a lack of in-depth research on the dosimetry, PDT is still unable to realize the precise killing of malignant tumors. Previous studies mainly focused on the mechanisms of one-parameter scaling in the killing effects of PDT, and they lack a full understanding of the synergistic effects of multiple parameters, which has led to many practical problems that could not be solved in clinical application. Therefore, in this study, we focused on the potential relationships among multiple PDT parameters and their impacts on the efficacy of PDT. By summarizing the precise killing law of sinoporphyrin sodium (DVDMS)-mediated PDT (DVDMS-PDT) on malignant tumor cells, we could promote the development of PDT dosimetry and clinical individualized precise treatment.

## 2. Results

### 2.1. Spectral Characteristics of DVDMS

In order to investigate the optimum cellular uptake in tumor cells, we first examined the absorption and fluorescence spectra of DVDMS. The spectral features of DVDMS were analyzed at different concentrations ranging from 6 to 12 µg/mL. As shown in Figure 1A, five different absorption peaks were observed at 379.5, 524.8, 554.3, 589.0 and 641.4 nm, and the maximum absorption wavelength was 379.5 nm. In addition, two fluorescence emission peaks were observed at 617.7 and 678.5 nm, of which the maximum wavelength was 617.7 nm (Figure 1B). These results provided references for the selection of the excitation and emission wavelengths in the subsequent experiments.

### 2.2. Investigation of Optimum Cellular Uptake of DVDMS

Then, we detected the accumulation of DVDMS in two types of malignant tumor cells after different incubation times. As shown in Figure 2A,C, after the treatment with 2 μg/mL DVDMS, the fluorescence intensity increased first, and then decreased in the tumor cells. To quantify the accumulation of DVDMS, the fluorescence intensity at different incubation time points was measured, and we found that the fluorescence intensity increased continuously to the maximum after incubation for 11 h in the U-118MG cells and 8 h in A375 cells, and began to decrease gradually in the following hours (Figure 2B,D). Therefore, we chose the optimum incubation time for DVDMS in the following experiments.

### 2.3. Identification of DVDMS-PDT Efficacy

First, we determined the toxicity threshold of DVDMS in two types of tumor cells. The cells were treated with different concentrations of DVDMS (0, 0.5, 1, 2, 4 and 8 μg/mL) for the optimum incubation time. As shown in Figure 3A,C, the survival rates of the U-118MG cells were 100.0%, 96.3%, 94.4%, 93.3%, 88.6% and 87.9%, and the survival rates of the A375 cells were 100.0%, 97.8%, 97.5%, 96.6%, 95.9% and 87.9%. The results indicated that the DVDMS concentrations ranging from 0 to 2 μg/mL had little toxicity to the cell viability in the two types of tumor cells. Therefore, we chose DVDMS concentrations lower than 2 µg/mL for the subsequent experiments. Then, we examined the killing effects of the DVDMS-PDT on two types of tumor cells. The results showed that neither light alone nor DVDMS alone showed significant killing effects on the cell viability (Figure 3B,D). However, after the treatment with the combination of DVDMS and light, the cell viabilities of the U-118MG and A375 cells were, respectively, only 26.4% and 6.0% (*p* < 0.001), suggesting the unique killing effects of DVDMS-PDT on malignant tumor cells.

### 2.4. In-Depth Analysis Associated Parameters with Outcomes in DVDMS-PDT

We investigated the relationships between the various parameters and PDT outcomes. Figure 4A,C shows that, under the same DVDMS concentration (2 μg/mL), the survival rate of the U-118MG and A375 cells decreased significantly with the increase in the irradiation time, indicating that there was probably a negative correlation between them. Similarly, the survival rate had the same trend as the power density. Therefore, we defined the product of the power density (*I*) and irradiation time (*t*) as the energy density (ED) (ED = *I* × *t*). As shown in Figure 4B,D, the survival rates of the U-118MG and A375 cells not only decreased with the increasing DVDMS concentration, but were also negatively correlated with the ED (*p* < 0.05). These results indicated that there might be a certain correlation among the power density, irradiation time and DVDMS concentration.

### 2.5. Establishment of Relationship among Parameters for PDT Efficacy

To further reveal the underlying relationship among the three parameters, we first defined the product of the power density (*I*), irradiation time (*t*) and DVDMS concentration (*c*) as the total photodynamic parameter (TPP) (TPP = *I* × *t* × *c*). As shown in Figure 5A,C, we found that the survival rates of the U-118MG and A375 cells were negatively correlated with the TPP value in the curve and were just an *e*-exponential function, which were, respectively, expressed as: *y* = 1.2*exp*(−0.1*x*) (*R*^2^ = 0.95, 1 ≤ *x* ≤ 23 J/cm^2^·μg/mL) and *y* = 1.2*exp*(−0.3*x*) (*R*^2^ = 0.97, 1 ≤ *x* ≤ 13 J/cm^2^·μg/mL). Moreover, the fitting accuracies of the formulas were both relatively high, with *R*^2^ (coefficients of determination) of 0.95 and 0.97, respectively, and the ranges of the TPP value were 1–23 J/cm^2^·μg/mL and 1–13 J/cm^2^·μg/mL, respectively. We also found that the results of the DVDMS-PDT did not conform to the *e*-exponential function if the TPP value was smaller (Figure 5A,C). Therefore, we selected multiple groups of TPPs (ranging from 0 to 1 J/cm^2^·μg/mL) to treat the cells to determine the value range of the *e*-exponential function. As shown in Figure 5B,D, when the TPP value was less than 1 J/cm^2^·μg/mL, the DVDMS-PDT had no killing effects on either the U-118MG or A375 cells. In addition, we further found that if the smallest dose of DVDMS was less than 0.5 µg/mL, then there were no PDT effects, regardless of the light dose (data not shown). Similarly, if the TPP value was higher than 23 J/cm^2^·μg/mL in the U-118MG cells or 13 J/cm^2^·μg/mL in the A375 cells, then the DVDMS-PDT was already in overtreatment for the tumor cells. Therefore, the formula had the value ranges and varied in different tumor-cell types.

### 2.6. Realization of Controllable Outcomes by Regulating Parameters of DVDMS-PDT

In order to further verify the accuracy and repeatability of the *e*-exponential function, we randomly selected three groups of TPPs (12.1, 6.0 and 3.9 J/cm^2^·μg/mL, according to the cell survival rates of 25%, 55% and 75% from the curve, respectively) to treat the U-118MG cells, and another three groups of TPPs (8.1, 3.4 and 1.8 J/cm^2^·μg/mL, according to the cell survival rates of 25%, 55% and 75% from the curve, respectively) to treat the A375 cells (Figure 6A,B). The results showed that the cell survival rates of the three groups in the U-118MG or A375 cells were well fit for the *e*-exponential function, demonstrating the accuracy of the *e*-exponential equation. Then, we chose nine different combinations of the power density, DVDMS concentration and irradiation time, and their product remained at 6.7 J/cm^2^·μg/mL in the U-118MG cells, and at 4.0 J/cm^2^·μg/mL in the A375 cells (i.e., the indicated TPP value of terminal half-cell death). Our data could perfectly match the expected results, indicating that, by using the fitted functional relationship, the treatment outcomes of the tumor cells could be controlled by adjusting the PDT parameters.

## 3. Discussion

Malignant tumors have always been the largest killers of humans worldwide [1]. Among them, glioblastoma (GBM) has the highest recurrence rate among all malignant tumors. The median progression-free survival (PFS) is 6.9 months, and the recurrence rate is close to 100% [19]. GBM belongs to diffuse astrocytic gliomas, which have no obvious boundary with normal brain tissue [20,21]. Even if combined with preoperative images and an intraoperative microscope, it is difficult to identify its accurate boundary, and the residual tumor cells after surgery can finally lead to recurrence [22]. Although expanding the extent of the resection can prolong the PFS and overall survival (OS) of patients with tumors, this strategy is not applicable to treating GBM [23]. Excessive resection means the loss of important neurological functions, which makes the patients unable to benefit. In addition, postoperative chemoradiotherapy has only a slight effect on the PFS and OS. Although TMZ in 2005, bevacizumab in 2009 and tumor treating fields (TTF) in 2015 have been approved by the FDA to treat recurrent GBM, the effects of these treatments are not significant, and the prognosis is still very poor [20]. Therefore, the precise killing of residual tumor cells is one of the major challenges faced by researchers.

PDT is one of the most successful optical applications in the biomedical field. It has significant outcomes in the treatment of various malignant tumors, and it is the first drug-instrument-combination technology approved by the FDA [4,5]. Compared with normal cells, the singlet oxygen produced by PDT tends to kill tumor cells, and it is not easy to produce drug resistance, which is very suitable for the treatment of GBM [3]. Photosensitizers (PSs) prefer to accumulate in GBM cells due to the existence of the blood–brain barrier. Studies have shown that the ratio of the PS concentration between GBM cells and normal cells is as high as 30:1, which is much higher than that of other tissues/organs [24]. In addition, the fluorescence characteristics of PSs themselves (i.e., cell labeling) are conductive to tumor resection or killing in the treatment of GBM [25,26]. Sinoporphyrin sodium (DVDMS) is a novel PS derived from Photofrin by Fang et al. [27]. Compared with the current clinical PSs, DVDMS had a good potential for clinical applications in PDT because of its higher chemical purity, better water solubility, better targeting and shorter skin-sensitivity period [28,29]. Our previous study showed that the singlet-oxygen quantum yield and the extinction coefficient of DVDMS at 630 nm were both significantly higher than that of Photofrin, leading to higher ^1^O_2_ production [30]. As a result, DVDMS is a promising PS for PDT [28,31]. Recently, DVDMS has entered the phase II clinical trial for the PDT treatment of advanced esophageal cancer patients (No. CTR20200598). Based on the above advantages, DVDMS was a good selection for PDT to treat malignant tumor cells in this study.

A number of clinical studies have fully demonstrated the effectiveness of PDT in the treatment of GBM [32,33,34,35,36]. In a meta-analysis of more than 1000 patients with GBM treated with intraoperative PDT, the median survival time of the patients increased up to 16.1 months, which was better than the standard treatment regimen [34]. Muller and Wilson reported that, on the basis of the same standard regimen of postoperative radiotherapy or chemotherapy, the median survival time of GBM patients in the additional adjuvant PDT groups was 3 months longer than that in the other groups, and the OS was also significantly increased [33]. These results show that PDT is one of the best treatment strategies for GBM. Similarly, more and more studies have also shown that PDT can be considered a reasonable option in the treatment of skin cancer, in which it has better efficacy (cure rates from 70% to 90%) and higher cosmetic outcomes versus surgery [37,38]. Even for “difficult-to-treat” lesions, PDT has excellent outcomes, with a 24-month complete response rate of 78% [39].

Although the effects of PDT in the treatment of malignant tumors are significant, it still cannot completely inhibit the recurrence of tumors. This is mainly because the current research on PDT dosimetry has not formed a system [40]. Researchers or surgeons prefer to use the empirical dose, which leads to two consequences: (1) the overtreatment of PDT, which can cause serious brain edema, resulting in neurological deterioration, and can even be life threatening; (2) the undertreatment of PDT, which cannot completely kill the residual tumor cells, which results in tumor recurrence [24,41]. Therefore, the key to PDT treatment is to precisely control the dose of PDT, killing the residual tumor cells while reducing edema.

As is known to all, the optical power density, irradiation time, PS concentration and oxygen concentration are important factors that affect the killing effects of PDT. Without the measurement and consideration of these parameters, PDT cannot be effectively implemented [15,42]. In order to precisely kill the residual tumor cells by adjusting the parameters, the mechanisms of various PDT parameters must be considered [25,43]. The main purpose of this study is to explore the internal relationship among the PDT parameters, and their impacts on PDT outcomes, so as to achieve the precise killing of residual tumor cells by regulating these parameters.

Considering the abundant blood vessels and sufficient oxygen supply in brain tissue or superficial tissue [44], we focused on three other PDT parameters: the optical power density (*I*), irradiation time (*t*) and PS concentration (*c*), and we defined their product as the total photodynamic parameter (TTP) (TTP = *I* × *t* × *c*). The correlation analysis showed that there was an *e*-exponential-function relationship among them. Furthermore, we proved the fitting accuracy and value range of the function. By randomly changing three parameters, the accuracy and repeatability of the function relationship were successfully verified.

This study revealed the underlying link among multiple PDT parameters by building a mathematical model, which has very important application values in the clinical treatment of malignant tumors. For example, by adjusting the power density or illumination time, the residual tumor cells can be killed synchronously, integrally and precisely according to the different concentration distributions of PSs in the tissues, thereby avoiding excess or insufficient treatment. Or, by appropriately enhancing the optical power density or illumination time to lower the dosage of PSs, it contributes to reducing the toxic side effects of PSs, easing the financial burden, and improving the quality of life in patients with malignant tumors. Or, by properly increasing the power density or PS concentration to reduce the illumination time of the PDT, it is helpful to decrease the exposure time of normal tissues and prevent infection.

## 4. Materials and Methods

### 4.1. Reagents

Sinoporphyrin sodium (DVDMS) (molecular formula: C_68_H_66_N_8_O_9_Na_4_; molecular weight: 1230.3; purity: >98%) was obtained from Qinglong High-Tech Co., Ltd. (Jiangxi, China). The DVDMS was dissolved in double-distilled water and stored in the dark at −20 °C. Figure 7 showed the chemical structure of DVDMS. The Cell Counting Kit-8 (CCK-8) was purchased from Beyotime Biotechnology (Shanghai, China).

### 4.2. Construction of PDT Device

As the light source, a diode laser (Xi’an Ningju Photoelectric Technology Co., Ltd., Xi’an, China) with a continuously variable output of 0–500 mW, and with a wavelength of 633 nm, was fixed on the bracket. The light was distributed by a 25 mm-diameter fiber collimator (Changchun New Industries Optoelectronics Technology Co., Ltd., Changchun, China) connected to the end of the diode laser to form a spot size of 14 mm in diameter. The 6.4 mm-diameter culture dish was placed on a speed-adjustable turntable to ensure that all the cells were uniformly irradiated (Figure 8). The laser output power was measured using a power meter (Ophir Photonics, Jerusalem, Israel).

### 4.3. Cell Line and Cell Culture

The human malignant glioblastoma U-118MG cell line and human melanoma A375 cell line were purchased from the Cell Bank of Type Culture Collection of Chinese Academy of Sciences (Shanghai, China). The cells were cultured in Dulbecco’s modified Eagle’s medium (DMEM) containing fetal bovine serum (10%), penicillin (100 μg/mL) and streptomycin (100 μg/mL), and they were then placed at 37 °C in a humidified incubator with 95% air and 5% CO_2_.

### 4.4. Spectral Characteristics Analysis

The UV–visible absorption spectra of DVDMS with a range of concentrations (6–12 μg/mL) were measured using a miniature fiber-optic spectrometer (QE65000, Ocean Optics Inc., Dunedin, FL, USA) with a deuterium lamp based on the Beer–Lambert law. The photoluminescence spectra of DVDMS were measured using a miniature fiber-optic spectrometer (USB4000, Ocean Optics Inc., Dunedin, FL, USA).

### 4.5. Cellular-Uptake Assay

The tumor cells (2 × 10^4^ cells per well) were cultured in serum-free DMEM in a 96-well plate and were treated with 2 μg/mL DVDMS at different time periods (0–12 h). Then, the cells were washed twice with cold phosphate-buffered saline (PBS), and the intracellular accumulation of DVDMS was confirmed by using FSX 100 fluorescence microscopy (Olympus, Tokyo, Japan). In order to quantify the cell uptake of DVDMS, the fluorescence was measured using a Spectramax M3 microplate reader (Molecular Devices, San Jose, CA, USA).

### 4.6. Cytotoxicity Assay

The DVDMS stock solution and DMEM medium were mixed to obtain the final solutions, with DVDMS concentrations of 0, 0.5, 1, 2, 4 and 8 μg/mL. The tumor cells (2 × 10^4^ cells per well) were cultured in serum-free DMEM in a 96-well plate for the indicated incubation time. The medium was then replaced by 100 μL of each concentration in triplicate. After the incubation, the cell viability was measured using the CCK-8 assay. In brief, the mixture of 90 μL of DMEM and 10 μL of CCK-8 was added in each well and incubated for 1 h at 37 °C. Then, the optical density (OD) of each well was immediately measured at 450 nm using a Spectramax M3 microplate reader.

### 4.7. PDT Treatment

The tumor cells (2 × 10^4^ cells per well) were cultured in serum-free DMEM in a 96-well plate for 24 h, and then incubated with different concentrations of DVDMS in the range of 0–2 μg/mL for the indicated time. The output power of the light was in the range of 0–110 mW, and the laser beam illuminated an area of 1.5 cm^2^, resulting in a power density in the range of 0–71.2 mW/cm^2^. The irradiation time was in the range of the indicated time. After the PDT treatment, the cell viability was measured using the CCK-8 assay. The data were fitted and calculated with GraphPad Prism 5.0 software (GraphPad Software Inc., La Jolla, CA, USA).

### 4.8. Statistical Analysis

All data were expressed as the means ± SDs based on at least three independent experiments. Statistical analysis was evaluated by a one-way ANOVA with Dunnett’s test using the GraphPad Prism 5.0 software. The values of * *p* < 0.05 and *** *p* < 0.001 were considered statistically significant.

## 5. Conclusions

In the current study, we first revealed the internal correlation among the parameters for PDT outcomes, and we realized the precision killing of malignant tumor cells by regulating the parameters in the DVDMS-PDT, which was conducive to the development of the PDT dosimetry and personalized precise treatment.

## Figures and Tables

**Figure 1 ijms-23-10561-f001:**
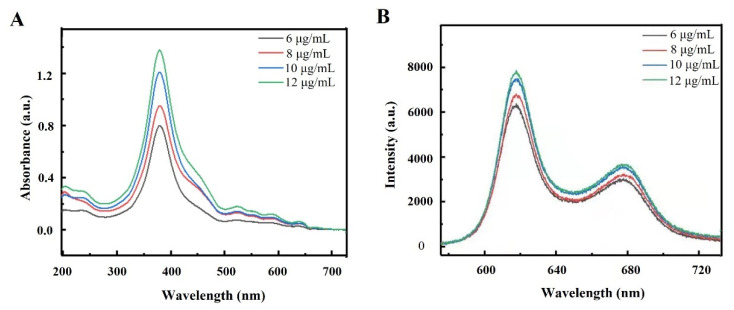
The spectral characteristics of DVDMS. The (**A**) absorption spectra and (**B**) emission spectra of DVDMS with different concentrations (6–12 µg/mL) were examined.

**Figure 2 ijms-23-10561-f002:**
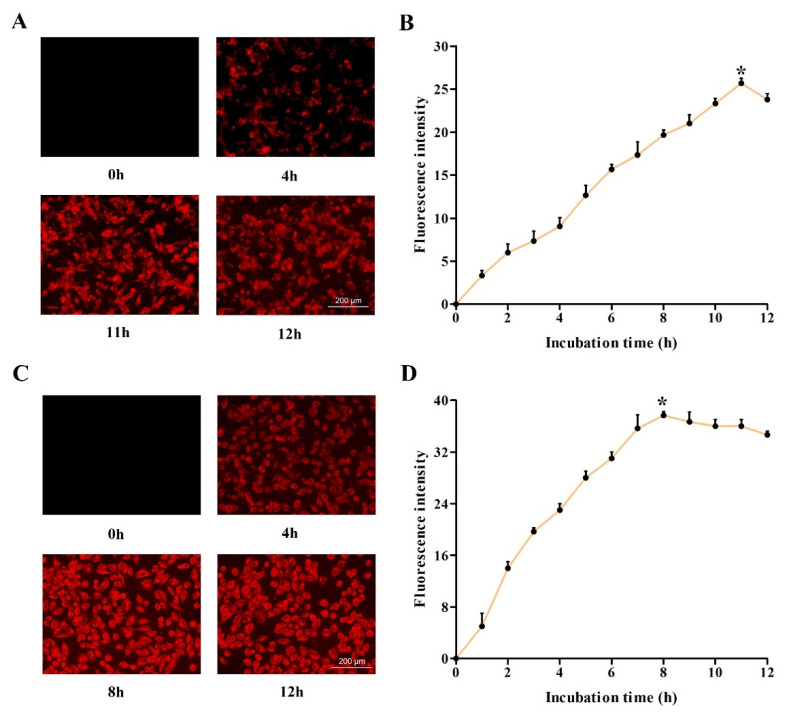
The determination of optimum incubation time for DVDMS. The cellular uptake of DVDMS (2 μg/mL) by fluorescence microscopy (magnification ×400) in (**A**) U-118MG cells and (**C**) A375 cells, and the fluorescence intensity of DVDMS by fluorescence plate reader in (**B**) U-118MG cells and (**D**) A375 cells. * *p* < 0.05 between groups. Data are expressed as mean ± SD of three independent experiments.

**Figure 3 ijms-23-10561-f003:**
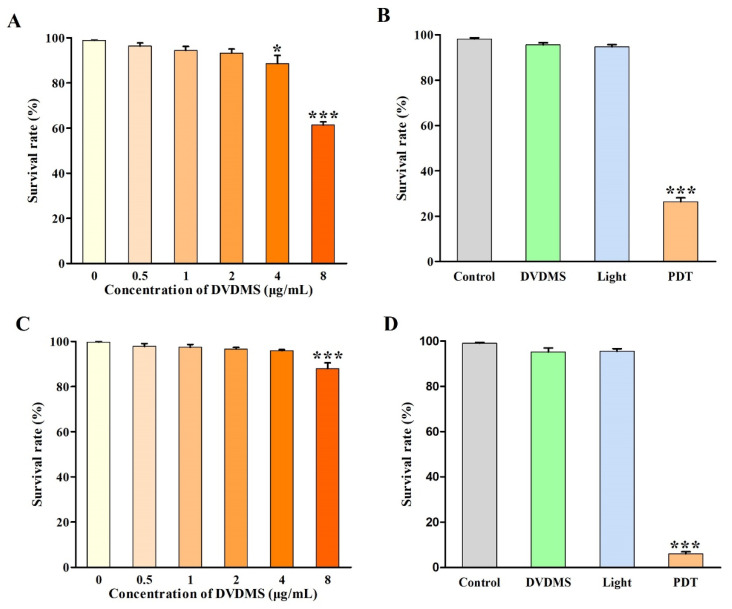
The identification of DVDMS-PDT efficacy. The cytotoxicity effects of DVDMS with different concentrations (0, 0.5, 1, 2, 4 and 8 µg/mL) for the optimum incubation time in (**A**) U-118MG cells and (**C**) A375 cells, and the phototoxicity effects of DVDMS (2 μg/mL) combined with light (62.6 mW/cm^2^) for 90 s in (**B**) U-118MG cells, and for 80 s in (**D**) A375 cells. * *p* < 0.05 and *** *p* < 0.001 vs. control group. Data are expressed as mean ± SD of three independent experiments.

**Figure 4 ijms-23-10561-f004:**
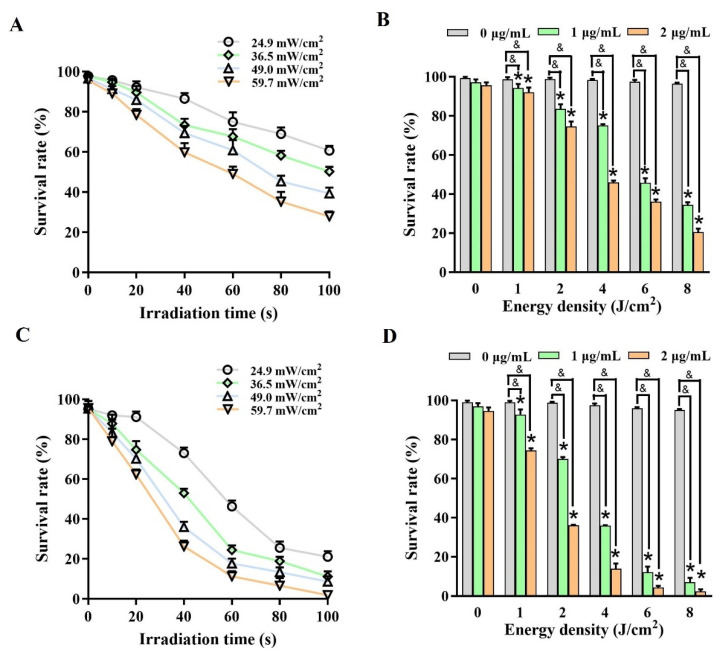
An analysis associated the parameters with the outcomes in DVDMS-PDT. The relationship of the power density (*I*) with the irradiation time (*t*) in the survival of (**A**) U-118MG cells and (**C**) A375 cells, and the relationship of the energy density (ED) (ED = *I* × *t*) with the DVDMS concentration (c) in the survival of (**B**) U-118MG cells and (**D**) A375 cells. * *p* < 0.05 vs. control group, and *p* < 0.05 between groups. Data are expressed as mean ± SD of three independent experiments.

**Figure 5 ijms-23-10561-f005:**
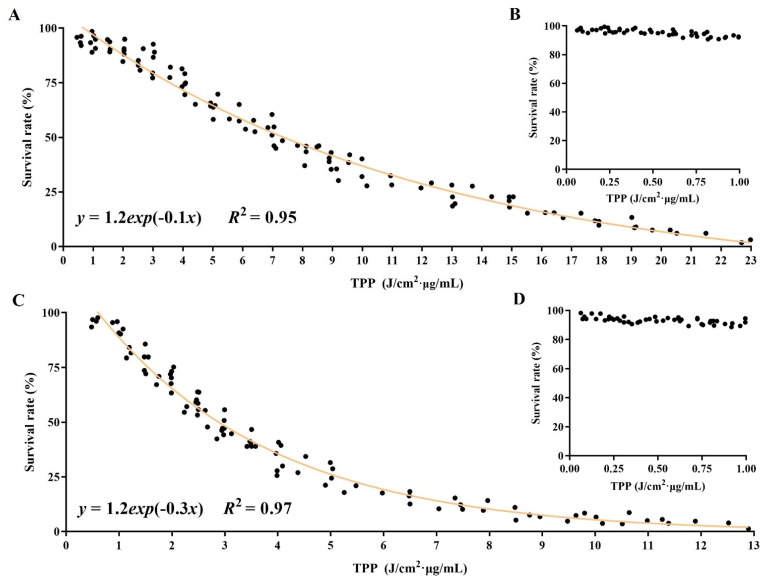
The establishment of the relationship among the DVDMS-PDT parameters. The *e*-exponential relationship with the total photodynamic parameters (TPP) in (**A**) U-118MG cells and (**C**) A375 cells. TPP = power density × irradiation time × DVDMS concentration (TPP = *I* × *t* × *c*). The changes in the survival of (**B**) U-118MG cells and (**D**) A375 cells, with TPP values ranging from 0 to 1 J/cm^2^·μg/mL.

**Figure 6 ijms-23-10561-f006:**
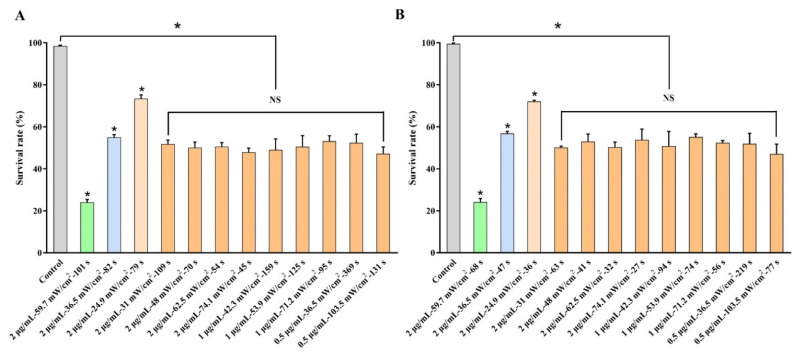
The realization of controllable outcomes by regulating the parameters of DVDMS-PDT. The expected killing effects of (**A**) U-118MG cells and (**B**) A375 cells were easily achieved according to the summarizing *e*-exponential formula. * *p* < 0.05 vs. control group, and NS with no significant difference between groups. Data are expressed as mean ± SD of three independent experiments.

**Figure 7 ijms-23-10561-f007:**
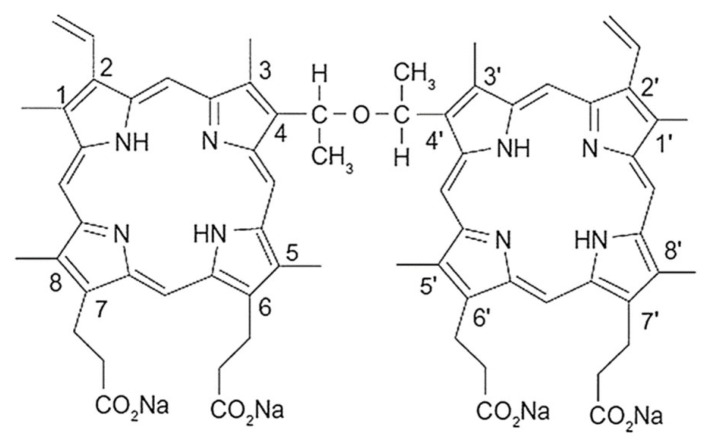
The chemical structure of DVDMS.

**Figure 8 ijms-23-10561-f008:**
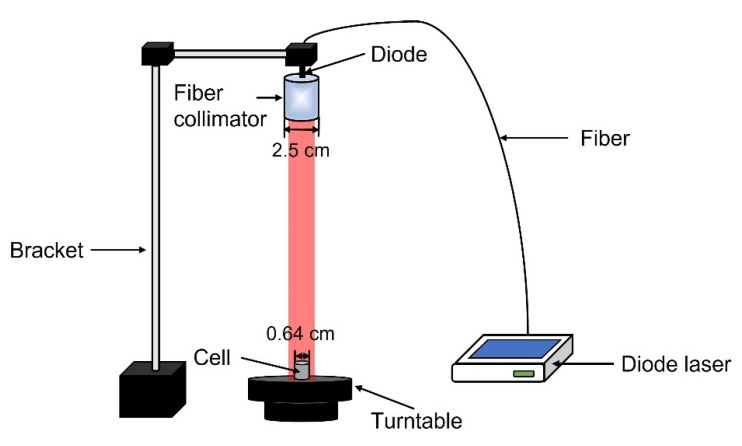
Schematic diagram of PDT device and experimental setup.

## Data Availability

The data that support the findings of this study are available from the corresponding author upon reasonable request.

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
