# Peer review of "Precision Killing of Sinoporphyrin Sodium-Mediated Photodynamic Therapy against Malignant Tumor Cells"

_ijms, 2022, doi:10.3390/ijms231810561_

Round 1

Reviewer 1 Report

The article is interesting and valuable. A few small clarifications: the scale in figure 2 should be indicated and a little more detailed in the conclusions with an emphasis on the future perspectives of the study.

Reviewer 2 Report

Guixiang et al. determine a mathematical formula to predict cell survival to DVDMS-PDT in a glioblastoma and a melanoma cell line which is very interesting. However, the lack of controls and the use of more cell lines to validate this formula is necessary before it can be published.

Major concerns:

1.      Figure 2 shows the fluorescence intensity of the DVDMS in the two cell lines indicating that "the fluorescence intensity increased firstly and then decreased in tumor cells". This is not what is seen in figure 2, where the fluorescence intensity increases until it reaches a maximum and then remains the same. Please correct this. Also, I think you should check the statistical differences as in figure 2D the times 9-11 are probably also statistically significant. Moreover, since it seems that the incubation in the following experiments is done for 24h, it would be interesting to show what happens to the fluorescence intensity at this time. Also, I suggest that they perform a flow cytometry assay to measure the DVDMS intensity at the different times since quantification by imaging is much less accurate.  

2.      In Figure 3 B and D, how are the irradiation times (90 and 80 s) selected?  Please indicate. Además deberían aclarar porqué incuban durante 24h en oscuridad con el DVDMS y no realizan la incubación en el pico de máxima intensidad de fluorescencia correspondiente de la figura 2.

3.      Figure 4a/c and b/d are the same. The measurement of energy or light power should be given in J/cm2 as this measurement is universal and each laboratory will be able to use the same dose taking into account the power of its lamp and calculating the irradiation time (W=J/t). Therefore, please change all figures to J/cm2 (not in time, s) and exclude figure 4a/c. Therefore, in your article you should indicate that cell survival is negatively dependent on concentration and energy density (J/cm2); TPP = ED x c.

4.      Finally, to prove the efficacy of your formulation, you should use healthy cell controls to confirm that they do not respond in the same way as tumor cells. In addition, you should use more glioblastoma and melanoma cell lines to confirm that various glioblastoma cells and various melanoma cells respond to the formulation. In addition, it would be interesting to know whether other photosensitizers also respond to the same formulation in these cell lines. This is very necessary because otherwise the article does not carry enough weight to be considered novel and to be able to try to translate it to the clinic as suggested by the authors. If each photosensitizer and cell type responds to a different mathematical formula, it is unfeasible for it to be easily applicable.

Minor concerns:

1.      Neither in material and methods nor in results is it clear how long the DVDMS is incubated in darkness before irradiation. Please indicate.

2.      Since the concentration selected is 2ug/ml, I consider that absorption and emission spectra at this concentration should be shown in figure 1, in addition to the higher concentrations.

Round 2

Reviewer 2 Report

Thank you very much for implementing the changes